# Few-Shot Segmentation via Cycle-Consistent Transformer

**Gengwei Zhang**[1,2*], **Guoliang Kang**[3], **Yi Yang**[4], **Yunchao Wei**[5,6†]

[1] Baidu Research
[2] ReLER, Centre for Artificial Intelligence, University of Technology Sydney
[3] University of Texas, Austin
[4] CCAI, College of Computer Science and Technology, Zhejiang University
[5] Institute of Information Science, Beijing Jiaotong University
[6] Beijing Key Laboratory of Advanced Information Science and Network
{zgwdavid, kgl.prml, wychao1987, yee.i.yang}@gmail.com

## Abstract

Few-shot segmentation aims to train a segmentation model that can fast adapt to novel classes with few exemplars. The conventional training paradigm is to learn to make predictions on query images conditioned on the features from support images. Previous methods only utilized the semantic-level prototypes of support images as the conditional information. These methods cannot utilize all pixel-wise support information for the query predictions, which is however critical for the segmentation task. In this paper, we focus on utilizing pixel-wise relationships between support and query images to facilitate the few-shot segmentation task. We design a novel **C**ycle-**C**onsistent **TR**ansformer (CyCTR) module to aggregate pixel-wise support features into query ones. CyCTR performs cross-attention between features from different images, *i.e.* support and query images. We observe that there may exist unexpected irrelevant pixel-level support features. Directly performing cross-attention may aggregate these features from support to query and bias the query features. Thus, we propose using a novel cycle-consistent attention mechanism to filter out possible harmful support features and encourage query features to attend to the most informative pixels from support images. Experiments on all few-shot segmentation benchmarks demonstrate that our proposed CyCTR leads to remarkable improvement compared to previous state-of-the-art methods. Specifically, on Pascal-$5^i$ and COCO-$20^i$ datasets, we achieve 67.5% and 45.6% mIoU for 5-shot segmentation, outperforming previous state-of-the-art method by 5.6% and 7.1% respectively.

## 1 Introduction

Recent years have witnessed great progress in semantic segmentation [19, 4, 47]. The success can be largely attributed to large amounts of annotated data [48, 17]. However, labeling dense segmentation masks are very time-consuming [45]. Semi-supervised segmentation [15, 39, 38] has been broadly explored to alleviate this problem, which assumes a large amount of unlabeled data is accessible. However, semi-supervised approaches may fail to generalize to novel classes with very few exemplars. In the extreme low data regime, few-shot segmentation [26, 35] is introduced to train a segmentation model that can quickly adapt to novel categories.

---

[*] Part of this work was done when Gengwei Zhang was an intern at Baidu Research.
[†] Corresponding author.

35th Conference on Neural Information Processing Systems (NeurIPS 2021).

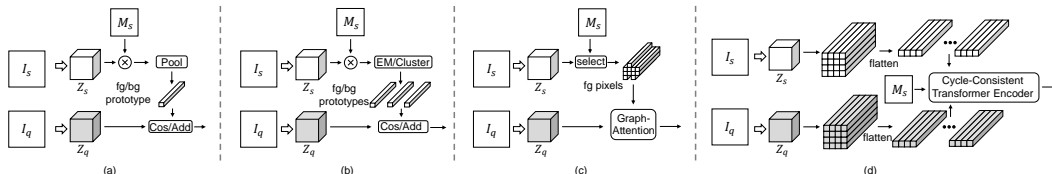

Figure 1: Different learning frameworks for few-shot segmentation, from the perspective of ways to utilize support information. (a) Class-wise mean pooling based method. (b) Clustering based method. (c) Foreground pixel attention method. (d) Our Cycle-Consistent TRansformer (CyCTR) framework that enables all beneficial support pixel-level features (foreground and background) to be considered.

Most few-shot segmentation methods follow a learning-to-learn paradigm where predictions of query images are made conditioned on the features and annotations of support images. The key to the success of this training paradigm lies in how to effectively utilize the information provided by support images. Previous approaches extract semantic-level prototypes from support features and follow a metric learning [29, 7, 35] pipeline extending from PrototypicalNet [28]. According to the granularity of utilizing support features, these methods can be categorized into two groups, as illustrated in Figure 1: 1) Class-wise mean pooling [35, 46, 44] (Figure 1(a)). Support features within regions of different categories are averaged to serve as prototypes to facilitate the classification of query pixels. 2) Clustering [18, 41] (Figure 1(b)). Recent works attempt to generate multiple prototypes via EM algorithm or K-means clustering [41, 18], in order to extract more abundant information from support images. These prototype-based methods need to "compress" support information into different prototypes (*i.e.* class-wise or cluster-wise), which may lead to various degrees of loss of beneficial support information and thus harm segmentation on query image. Rather than using prototypes to abstract the support information, [43, 34] (Figure 1(c)) propose to employ the attention mechanism to extract information from support foreground pixels for segmenting query. However, such methods ignore all the background support pixels that can be beneficial for segmenting query image, and incorrectly consider partial foreground support pixels that are quite different from the query ones, leading to sub-optimal results.

In this paper, we focus on equipping each query pixel with relevant information from support images to facilitate the query pixel classification. Inspired by the transformer architecture [32] which performs feature aggregation through attention, we design a novel **C**ycle-**C**onsistent **Tr**ansformer (CyCTR) module (Figure 1(d)) to aggregate pixel-wise support features into query ones. Specifically, our CyCTR consists of two types of transformer blocks: the self-alignment block and the cross-alignment block. The self-alignment block is employed to encode the query image features by aggregating its relevant context information, while the cross-alignment aims to aggregate the pixel-wise features of support images into the pixel-wise features of query image. Different from self-alignment where Query[3], Key and Value come from the same embedding, cross-alignment takes features from query images as Query, and those from support images as Key and Value. In this way, CyCTR provides abundant pixel-wise support information for pixel-wise features of query images to make predictions.

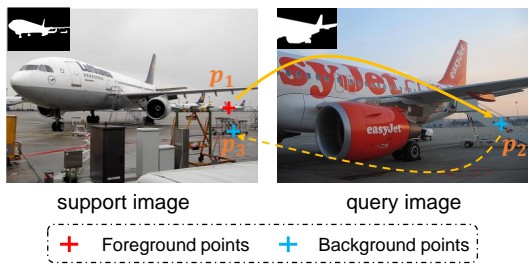

support image        query image

+ Foreground points        + Background points

Figure 2: The motivation of our proposed method. Many pixel-level support features are quite different from the query ones, and thus may confuse the attention. We incorporate cycle-consistency into attention to filter such confusing support features. Note that the confusing support features may come from foreground and background.

Moreover, we observe that due to the differences between support and query images, *e.g.*, scale, color and scene, only a small proportion of support pixels can be beneficial for the segmentation of query image. In other words, in the support image, some pixel-level information may confuse the attention in the transformer. Figure 2 provides a visual example of a support-query pair together with the label

---

[3]To distinguish from the phrase "query" in few-shot segmentation, we use "Query" with capitalization to note the query sequence in the transformer.

masks. The confusing support pixels may come from both foreground pixels and background pixels. For instance, point $p_1$ in the support image located in the plane afar, which is indicated as foreground by the support mask. However, the nearest point $p_2$ in the query image (*i.e.* $p_2$ has the largest feature similarity with $p_1$) belongs to a different category, *i.e.* background. That means, there exists no query pixel which has both high similarity and the same semantic label with $p_1$. Thus, $p_1$ is likely to be harmful for segmenting "plane" and should be ignored when performing the attention. To overcome this issue, in CyCTR, we propose to equip the cross-alignment block with a novel cycle-consistent attention operation. Specifically, as shown in Figure 2, starting from the feature of one support pixel, we find its nearest neighbor in the query features. In turn, this nearest neighbor finds the most similar support feature. If the starting and the end support features come from the same category, a cycle-consistency relationship is established. We incorporate such an operation into attention to force query features only attend to cycle-consistent support features to extract information. In this way, the support pixels that are far away from query ones are not considered. Meanwhile, cycle-consistent attention enables us to more safely utilize the information from background support pixels, without introducing much bias into the query features.

In a nutshell, our contributions are summarized as follows: (1) We tackle few-shot segmentation from the perspective of providing each query pixel with relevant information from support images through pixel-wise alignment. (2) We propose a novel Cycle-Consistent TRansformer (CyCTR) to aggregate the pixel-wise support features into the query ones. In CyCTR, we observe that many support features may confuse the attention and bias pixel-level feature aggregation, and propose incorporating cycle-consistent operation into the attention to deal with this issue. (3) Our CyCTR achieves state-of-the-art results on two few-shot segmentation benchmarks, *i.e.*, Pascal-$5^i$ and COCO-$20^i$. Extensive experiments validate the effectiveness of each component in our CyCTR.

## 2 Related Work

### 2.1 Few-Shot Segmentation

Few-shot segmentation [26] is established to perform segmentation with very few exemplars. Recent approaches formulate few-shot segmentation from the view of metric learning [29, 7, 35]. For instance, [7] first extends PrototypicalNet [28] to perform few-shot segmentation. PANet [35] simplifies the framework with an efficient prototype learning framework. SG-One [46] leverage the cosine similarity map between the single support prototype and query features to guide the prediction. CANet [44] replaces the cosine similarity with an additive alignment module and iteratively refines the network output. PFENet [30] further designs an effective feature pyramid module and leverages a prior map to achieve better segmentation performance. Recently, [41, 18, 43] point out that only a single support prototype is insufficient to represent a given category. Therefore, they attempt to obtain multiple prototypes via EM algorithm to represent the support objects and the prototypes are compared with query image based on cosine similarity [18, 41]. Besides, [43, 34] attempt to use graph attention networks [33, 40] to utilize all foreground support pixel features. However, they ignore all pixels in the background region by default. Besides, due to the large difference between support and query images, not all support pixels will benefit final query segmentation. Recently, some concurrent works propose to learn dense matching through Hypercorrelation Squeeze Networks [22] or mining latent classes [42] from the background region. Our work aims at mining information from the whole support image, but exploring to use the transformer architecture and from a different perspective, *i.e.,* reducing the noise in the support pixel-level features.

### 2.2 Transformer

Transformer and self-attention were firstly introduced in the fields of machine translation and natural language processing [6, 32], and are receiving increasing interests recently in the computer vision area. Previous works utilize self-attention as additional module on top of existing convolutional networks, *e.g.,* Nonlocal [36] and CCNet [14]. ViT [8] and its following work [31] demonstrate the pure transformer architecture can achieve state-of-the-art for image recognition. On the other hand, DETR [3] builds up an end-to-end framework with a transformer encoder-decoder on top of backbone networks for object detection. And its deformable vairents [51] improves the performance and training efficiency. Besides, in natural language processing, a few works [2, 5, 27] have been

introduced for long documents processing with sparse transformers. In these works, each Query token only attends to a pre-defined subset of Key positions.

## 2.3 Cycle-consistency Learning

Our work is partially inspired by cycle-consistency learning [50, 9] that is explored in various computer vision areas. For instance, in image translation, CycleGAN [50] uses cycle-consistency to align image pairs. It is also effective in learning 3D correspondence [49], consistency between video frames [37] and association between different domains [16]. These works typically constructs cycle-consistency loss between aligned targets (*e.g.*, images). However, the simple training loss cannot be directly applied to few-shot segmentation because the test categories are unseen from the training process and no finetuning is involved during testing. In this work, we incorporate the idea of cycle-consistency into transformer to eliminate the negative effect of confusing or irrelevant support pixels.

# 3 Methodology

## 3.1 Problem Setting

Few-shot segmentation aims at training a segmentation model that can segment novel objects with very few annotated samples. Specifically, given dataset $D_{train}$ and $D_{test}$ with category set $C_{train}$ and $C_{test}$ respectively, where $C_{train} \cap C_{test} = \emptyset$, the model trained on $D_{train}$ is directly used to test on $D_{test}$. In line with previous works [30, 35, 44], episode training is adopted in this work for few-shot segmentation. Each episode is composed of $k$ support images $I_s$ and a query image $I_q$ to form a $k$-shot episode $\{\{I_s\}^k, I_q\}$, in which all $\{I_s\}^k$ and $I_q$ contain objects from the same category. Then the training set and test set are represented by $D_{train} = \{\{I_s\}^k, I_q\}^{N_{train}}$ and $D_{test} = \{\{I_s\}^k, I_q\}^{N_{test}}$, where $N_{train}$ and $N_{test}$ is the number of episodes for training and test set. During training, both support masks $M_s$ and query masks $M_q$ are available for training images, and only support masks are accessible during testing.

## 3.2 Revisiting of Transformer

Following the general form in [32], a transformer block is composed of alternating layers of multi-head attention (MHA) and multi-layer perceptron (MLP). LayerNorm (LN) [1] and residual connection [12] are applied at the end of each block. Specially, an attention layer is formulated as

$$\text{Atten}(Q, K, V) = \text{softmax}(\frac{QK^T}{\sqrt{d}})V, \tag{1}$$

where $[Q; K; V] = [W_q Z_q; W_k Z_{kv}; W_v Z_{kv}]$, in which $Z_q$ is the input Query sequence, $Z_{kv}$ is the input Key/Value sequence, $W_q, W_k, W_v \in \mathbb{R}^{d \times d}$ denote the learnable parameters, $d$ is the hidden dimension of the input sequences and we assume all sequences have the same dimension $d$ by default. For each Query element, the attention layer computes its similarities with all Key elements. Then the computed similarities are normalized via $\text{softmax}$, which are used to multiply the Value elements to achieve the aggregated outputs. When $Z_q = Z_{kv}$, it functions as self-attention mechanism.

The multi-head attention layer is an extention of attention layer, which performs $h$ attention operations and concatenates consequences together. Specifically,

$$\text{MHA}(Q, K, V) = [\text{head}_1, ..., \text{head}_h], \tag{2}$$

where $\text{head}_m = \text{Atten}(Q_m, K_m, V_m)$ and the inputs $[Q_m, K_m, V_m]$ are the $m^{th}$ group from $[Q, K, V]$ with dimension $d/h$.

## 3.3 Cycle-Consistent Transformer

Our framework is illustrated in Figure 3(a). Generally, an encoder of our Cycle-Consistent TRansformer (CyCTR) consists of a self-alignment transformer block for encoding the query features and a cross-alignment transformer block to enable the query features to attend to the informative support features. The whole CyCTR module stacks $L$ encoders.

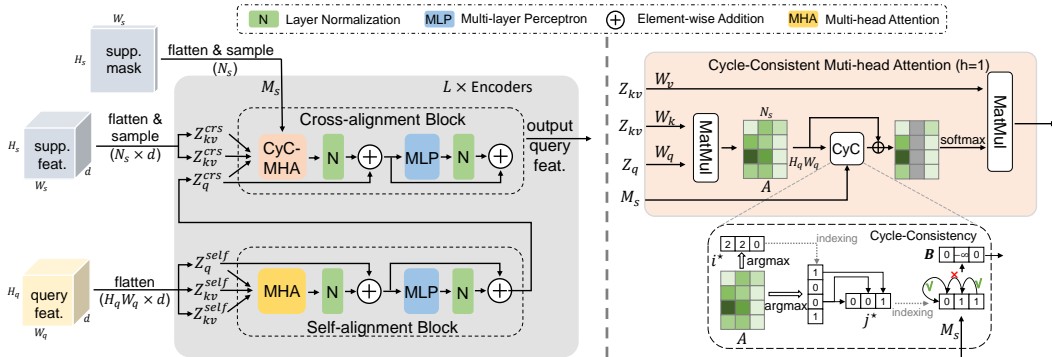

Figure 3: Framework of our proposed Cycle-Consistent TRansformer (CyCTR). Each encoder of CyCTR consists of two transformers blocks, *i.e.*, the self-alignment block for utilizing global context within the query feature map and the cross-alignment block for aggregate information from support images. In the cross-alignment block, we introduce the multi-head cycle-consistent attention (shown on the right, with the number of heads $h = 1$ for simplicity). The attention operation is guided by the cycle-consistency among query and support features.

Specifically, for the given query feature $X_q \in \mathbb{R}^{H_q \times W_q \times d}$ and support feature $X_s \in \mathbb{R}^{H_s \times W_s \times d}$, we first flatten them into 1D sequences (with shape $HW \times d$) as inputs for transformer, in which a *token* is represented by the feature $z \in \mathbb{R}^d$ at one pixel location. The self-alignment block only takes the flattened query feature as input. As context information of each pixel has been proved beneficial for segmentation [4, 47], we adopt the self-alignment block to pixel-wise features of query image to aggregate their global context information. We don't pass support images through the self-alignment block, as we mainly focus on the segmentation performance of query images. Passing through the support images which don't coordinate with the query mask may do harm to the self-alignment on query images.

In contrast, the cross-alignment block performs attention between query and support pixel-wise features to aggregate relevant support features into query ones. It takes the flattened query feature and a subset of support feature (the sampling procedure is discussed latter) with size $N_s \leq H_s W_s$ as Key/Value sequence $Z_{kv}$.

With these two blocks, it is expected to better encoder the query features to facilitate the subsequent pixel-wise classification. When stacking $L$ encoders, the output of the previous encoder is fed into the self-alignment block. The outputs of self-alignment block and the sampled support features are then fed into the cross-alignment block.

### 3.3.1 Cycle-Consistent Attention

According to the aforementioned discussion, the pure pixel-level attention may be confused by excessive irrelevant support features. To alleviate this issue, as shown in Figure 3(b), a cycle-consistent attention operation is proposed. We first go through the proposed approach for 1-shot case for presentation simplicity and then discuss it in the multiple shot setting.

Formally, an affinity map $A = \frac{QK^T}{\sqrt{d}}, A \in \mathbb{R}^{H_q W_q \times N_s}$ is first calculated to measure the correspondence between all query and support pixels. Then, for an arbitrary support pixel/token $j$ ($j \in \{0, 1, ..., N_s - 1\}$, $N_s$ is the number of support pixels), its most similar query pixel/token $i^\star$ is obtained by

$$i^\star = \underset{i}{\operatorname{argmax}} A_{(i,j)}, \tag{3}$$

where $i \in \{0, 1, ..., H_q W_q - 1\}$ denotes the spatial index of query pixels. Since the query mask is not accessible, the label of query pixel $i^\star$ is unknown. However, we can in turn find its most similar support pixel $j^\star$ in the same way:

$$j^\star = \underset{j}{\operatorname{argmax}} A_{(i^\star,j)}. \tag{4}$$

Given the sampled support label $M_s \in \mathbb{R}^{N_s}$, cycle-consistency is satisfied if $M_{s(j)} = M_{s(j^\star)}$. Previous work [16] attempts to encourage the feature similarity between cycle-consistent pixels to improve the model's generalization ability within the same set of categories. However, in few-shot segmentation, the goal is to enable the model to fast adapt to novel categories rather than making the model fit better to training categories. Thus, we incorporate the cycle-consistency into the attention operation to encourage the cycle-consistent cross-attention. First, by traversing all support tokens, an additive bias $B \in \mathbb{R}^{N_s}$ is obtained by

$$ B_j = \left\{ \begin{array}{ll} 0, & \text{if} M_{s(j)} = M_{s(j^\star)} \\ -\infty, & \text{if} M_{s(j)} \neq M_{s(j^\star)} \end{array} \right. , $$

where $j \in \{0, 1, ..., N_s\}$. Then, for a single query token $Z_{q(i)} \in \mathbb{R}^d$ at location $i$, the support information is aggregated by

$$ \text{CyCAtten}(Q_i, K_i, V_i) = \text{softmax}(A_{(i)} + B)V, \qquad (5) $$

where $i \in \{0, 1, ..., H_q W_q\}$ and $A$ is obtained by $\frac{QK^T}{\sqrt{d}}$. In the forward process, $B$ is element-wise added with the affinity $A_{(i)}$ for $Z_{q(i)}$ to aggregate support features. In this way, the attention weight for the cycle-inconsistent support features become zero, implying that these irrelevant information will not be considered. Besides, the cycle-consistent attention implicitly encourages the consistency between the most relevant query and support pixel-wise features through backpropagation. Note that our method aims at removing support pixels with certain inconsistency, rather than ensuring all support pixels to form cycle-consistency, which is impossible without knowing the query ground truth labels.

When performing self-attention in the self-alignment block, there may also exist the same issue, *i.e.* the query token may attend to irrelevant or even harmful features (especially when background is complex). According to our cycle-consistent attention, each query token should receive information from more consistent pixels than aggregating from all pixels. Due to the lack of query mask $M_q$, it is impossible to establish the cycle-consistency among query pixels/tokens. Inspired by DeformableAttention [51], the consistent pixels can be obtained via a learnable way as $\Delta = f(Q + \text{Coord})$ and $A^{'} = g(Q + \text{Coord})$, where $\Delta \in \mathbb{R}^{H_p W_p \times P}$ is the predicted consistent pixels, in which each element $\delta \in \mathbb{R}^P$ in $\Delta$ represents the relative offset from each pixel and $P$ represents the number of pixels to aggregate. And $A^{'} \in \mathbb{R}^{H_q W_q \times P}$ is the attention weights. $\text{Coord} \in \mathbb{R}^{H_q W_q \times d}$ is the positional encoding [24] to make the prediction be aware of absolute position, and $f(\cdot)$ and $g(\cdot)$ are two fully connected layers that predict the offsets[4] and attention weights. Therefore, the self-attention within the self-alignment transformer block is represented as

$$ \text{PredAtten}(Q_r, V_r) = \sum_{g}^{P} \text{softmax}(A^{'})_{(r,g)} V_{r+\Delta_{(r,g)}}, \qquad (6) $$

where $r \in \{0, 1, ..., H_q W_q\}$ is the index of the flattened query feature, both $Q$ and $V$ are obtained by multiplying the flattened query feature with the learnable parameter.

Generally speaking, the cycle-consistent transformer effectively avoids the attention being biased by irrelevant features to benefit the training of few-shot segmentation.

**Mask-guided sparse sampling and $K$-shot Setting**: Our proposed cycle-consistency transformer can be easily extended to $K$-shot setting where $K > 1$. When multiple support feature maps are provided, all support features are flattened and concatenated together as input. As the attention is performed at the pixel-level, the computation load will be high if the number of support pixels/tokens is large, which is usually the case under $K$-shot setting. In this work, we apply a simple mask-guided sampling strategy to reduce the computation complexity and make our method more scalable. Concretely, given the $k$-shot support sequence $Z_s \in \mathbb{R}^{kH_s W_s \times d}$ and the flattened support masks $M_s \in \mathbb{R}^{kH_s W_s}$, the support pixels/tokens are obtained by uniformly sampling $N_{fg}$ tokens ($N_{fg} <= \frac{N_s}{2}$, where $N_s \leq kH_s W_s$) from the foreground regions and $N_s - N_{fg}$ tokens from the background regions in all support images. With a proper $N_s$, the sampling operation reduces the computational complexity, and makes our algorithm more scalable with the increase of spatial size of support images. Additionally, this strategy helps balance the foreground-background ratio and also implicitly considers different sizes of various object regions in support images.

---

[4]The offsets are predicted as 2d coordinates and transformed into 1d coordinates.

Table 1: Comparison with other state-of-the-art methods for 1-shot and 5-shot segmentation on PASCAL-$5^i$ using the mIoU (%) evaluation metric. Best results are shown in bold.

| Method | Backbone | 1-shot | | | | | 5-shot | | | | |
|---|---|---|---|---|---|---|---|---|---|---|---|
| | | $5^0$ | $5^1$ | $5^2$ | $5^3$ | Mean | $5^0$ | $5^1$ | $5^2$ | $5^3$ | Mean |
| PANet [35] | Vgg-16 | 42.3 | 58.0 | 51.1 | 41.2 | 48.1 | 51.8 | 64.6 | 59.8 | 46.5 | 55.7 |
| FWB [23] | | 47.0 | 59.6 | 52.6 | 48.3 | 51.9 | 50.9 | 62.9 | 56.5 | 50.1 | 55.1 |
| SG-One [46] | | 40.2 | 58.4 | 48.4 | 38.4 | 46.3 | 41.9 | 58.6 | 48.6 | 39.4 | 47.1 |
| RPMM [41] | | 47.1 | 65.8 | 50.6 | 48.5 | 53.0 | 50.0 | 66.5 | 51.9 | 47.6 | 54.0 |
| CANet [44] | Res-50 | 52.5 | 65.9 | 51.3 | 51.9 | 55.4 | 55.5 | 67.8 | 51.9 | 53.2 | 57.1 |
| PGNet [43] | | 56.0 | 66.9 | 50.6 | 50.4 | 56.0 | 57.7 | 68.7 | 52.9 | 54.6 | 58.5 |
| RPMM [41] | | 55.2 | 66.9 | 52.6 | 50.7 | 56.3 | 56.3 | 67.3 | 54.5 | 51.0 | 57.3 |
| PPNet [18] | | 47.8 | 58.8 | 53.8 | 45.6 | 51.5 | 58.4 | 67.8 | **64.9** | 56.7 | 62.0 |
| PFENet [30] | | 61.7 | 69.5 | 55.4 | 56.3 | 60.8 | 63.1 | 70.7 | 55.8 | 57.9 | 61.9 |
| CyCTR (Ours) | Res-50 | **65.7** | **71.0** | **59.5** | **59.7** | **64.0** | **69.3** | **73.5** | 63.8 | **63.5** | **67.5** |
| FWB [23] | Res-101 | 51.3 | 64.5 | 56.7 | 52.2 | 56.2 | 54.9 | 67.4 | **62.2** | 55.3 | 59.9 |
| DAN [34] | | 54.7 | 68.6 | **57.8** | 51.6 | 58.2 | 57.9 | 69.0 | 60.1 | 54.9 | 60.5 |
| PFENet [30] | | 60.5 | 69.4 | 54.4 | 55.9 | 60.1 | 62.8 | 70.4 | 54.9 | 57.6 | 61.4 |
| CyCTR (Ours) | Res-101 | **67.2** | **71.1** | 57.6 | **59.0** | **63.7** | **71.0** | **75.0** | 58.5 | **65.0** | **67.4** |

## 3.4 Overall Framework

Following previous works [30, 35, 44], both query and support images are first feed into a shared backbone (*e.g.,* ResNet [12]) which is initialized with weights pretrained from ImageNet [25] to obtain general image features. Similar to [30], middle-level query features (the concatenation of query features from the $3^{rd}$ and the $4^{th}$ blocks of ResNet) are processed by a $1\times1$ convolution to reduce the hidden dimension. The high-level query features (from the $5^{th}$ block) are used to generate a prior map (the prior map is generated by calculating the pixel-wise similarity between query and support features, details can be found in the supplementary materials) and then are concatenated with the middle-level query features. The average masked support feature is also concatenated to provide global support information. The concatenated features are processed by a $1\times1$ convolution. The output query features are then fed into our proposed CyCTR encoders. The output of CyCTR encoders is fed into a classifier to obtain the final segmentation results. The classifier consists of a $3\times3$ convolutional layer, a ReLU layer and a $1\times1$ convolutional layer. More details about our network structure can be found in the supplementary materials.

# 4 Experiments

## 4.1 Dataset and Evaluation Metric

We conduct experiments on two commonly used few-shot segmentation datasets, Pascal-$5^i$ [10] (which is combined with SBD [11] dataset) and COCO-$20^i$ [17], to evaluate our method. For Pascal-$5^i$, 20 classes are separated into 4 splits. For each split, 15 classes are used for training and 5 classes for test. At the test time, 1,000 pairs that belong to the testing classes are sampled from the validation set for evaluation. In COCO-$20^i$, we follow the data split settings in FWB [23] to divide 80 classes evenly into 4 splits, 60 classes for training and test on 20 classes, and 5,000 validation pairs from the 20 classes are sampled for evaluation. Detailed data split settings can be found in the supplementary materials. Following common practice [30, 35, 46], the mean intersection over union (mIoU) is adopted as the evaluation metric, which is the averaged value of IoU of all test classes. We also report the foreground-background IoU (FB-IoU) for comparison.

## 4.2 Implementation Details

In our experiments, the training strategies follow the same setting in [30]: training for 50 epochs on COCO-$20^i$ and 200 epochs on Pascal-$5^i$. Images are resized and cropped to $473 \times 473$ for both datasets and we use random rotation from $-10°$ to $10°$ as data augmentation. Besides, we use ImageNet [25] pretrained ResNet [12] as the backbone network and its parameters (including BatchNorms) are frozen. For the parameters except those in the transformer layers, we use the initial learning rate $2.5 \times 10^{-3}$, momentum 0.9, weight decay $1 \times 10^{-4}$ and SGD optimizer with

Table 2: Comparison with other state-of-the-art methods for 1-shot and 5-shot segmentation on COCO-$20^i$ using the mIoU (%) evaluation metric. Best results are shown in bold.

| Method | Backbone | 1-shot | | | | | 5-shot | | | | |
|---|---|---|---|---|---|---|---|---|---|---|---|
| | | $20^0$ | $20^1$ | $20^2$ | $20^3$ | Mean | $20^0$ | $20^1$ | $20^2$ | $20^3$ | Mean |
| FWB [23] | Res-101 | 19.9 | 18.0 | 21.0 | 28.9 | 21.2 | 19.1 | 21.5 | 23.9 | 30.1 | 23.7 |
| PPNet [18] | Res-50 | 28.1 | 30.8 | 29.5 | 27.7 | 29.0 | 39.0 | 40.8 | 37.1 | 37.3 | 38.5 |
| RPMM [41] | Res-50 | 29.5 | 36.8 | 29.0 | 27.0 | 30.6 | 33.8 | 42.0 | 33.0 | 33.3 | 35.5 |
| PFENet [30] | Res-101 | 34.3 | 33.0 | 32.3 | 30.1 | 32.4 | 38.5 | 38.6 | 38.2 | 34.3 | 37.4 |
| CyCTR (Ours) | Res-50 | **38.9** | **43.0** | **39.6** | **39.8** | **40.3** | **41.1** | **48.9** | **45.2** | **47.0** | **45.6** |

poly learning rate decay [4]. The mini batch size on each gpu is set to 4. Experiments are carried out on Tesla V100 GPUs. For Pascal-$5^i$, one model is trained on a single GPU, while for COCO-$20^i$, one model is trained with 4 GPUs. We construct our baseline as follows: as stated in Section 3.4, the middle-level query features from backbone network are concatenated and merged with the global support feature and the prior map. This feature is processed by two residule blocks and input to the same classifier as our method. Dice loss [21] is used as the training objective. Besides, the middle-level query feature is averaged using the ground truth and concatenated with support feature to predict the support segmentation map, which produces an auxiliary loss for aligning features. The same settings are also used in our method except that we use our cycle-consistent transformer to process features rather than the residule blocks. For the proposed cycle-consistent transformer, we set the number of sampled support tokens $N_s$ to 600 for 1-shot and $5 \times 600$ for 5-shot setting. The number of sampled tokens is obtained according to the averaged number of foreground pixels among Pascal-$5^i$ training set. For the self-attention block, the number of points $P$ is set to 9. For other hyper-parameters in transformer blocks, we use $L = 2$ transformer encoders. We set the hidden dimension of MLP layer to $3 \times 256$ and that of input to 256. The number of heads for all attention layers is set to 8 for Pascal-$5^i$ and 1 for COCO-$20^i$. Parameters in the transformer blocks are optimized with AdamW [20] optimizer following other transformer works [3, 8, 31], with learning rate $1 \times 10^{-4}$ and weight decay $1 \times 10^{-2}$. Besides, we use Dropout with the probability 0.1 in all attention layers.

## 4.3 Comparisons with State-of-the-Art Methods

In Table 1 and Table 2, we compare our method with other state-of-the-art few-shot segmentation approaches on Pascal-$5^i$ and COCO-$20^i$ respectively. It can be seen that our approach achieves new state-of-the-art performance on both Pascal-$5^i$ and COCO-$20^i$. Specifically, on Pascal-$5^i$, to make fair comparisons with other methods, we report results with both ResNet-50 and ResNet-101. Our CyCTR achieves 64.0% mIoU with ResNet-50 backbone and 63.7% mIoU with ResNet-101 backbone for 1-shot segmentation, significantly outperforming previous state-of-the-art results by 3.2% and 3.6%, respectively. For 5-shot segmentation, our CyCTR can even surpass state-of-the art methods by 5.6% and 6.0% mIoU when using ResNet-50 and ResNet-

Table 3: Comparison with other methods using FB-IoU (%) on Pascal-$5^i$ for 1-shot and 5-shot segmentation.

| Method | Backbone | FB-IoU (%) | |
|---|---|---|---|
| | | 1-shot | 5-shot |
| A-MCG [13] | Res-101 | 61.2 | 62.2 |
| DAN [34] | Res-101 | 71.9 | 72.3 |
| PFENet [30] | Res-101 | 72.9 | 73.5 |
| CyCTR (Ours) | Res-101 | 73.0 | **75.4** |

101 backbones respectively. For COCO-$20^i$ results in Table 2, our method also outperforms other methods by a large margin due to the capability of the transformer to fit more complex data. Besides, Table 3 shows the comparison using FB-IoU on PASCAL-$5^i$ for 1-shot and 5-shot segmentation, our method also obtains the state-of-the-art performance.

## 4.4 Ablation Studies

To provide a deeper understanding of our proposed method, we show ablation studies in this section. The experiments are performed on Pascal-$5^i$ 1-shot setting with ResNet-50 as the backbone network, and results are reported in terms of mIoU.

Table 4: Ablation studies that validate the effectiveness of each component in our Cycle-Consistent TRansformer. The first result is obtained by our baseline (see Section 4.2 for details).

| self-alignment | cross-alignment | CyCTR (pred) | CyCTR (fg. only) | CyCTR | mIoU (%) |
|:---:|:---:|:---:|:---:|:---:|:---:|
|  |  |  |  |  | 59.3 |
| ✓ |  |  |  |  | 62.5 |
| ✓ | ✓ |  |  |  | 62.9 |
| ✓ | ✓ | ✓ |  |  | 62.6 |
| ✓ | ✓ |  | ✓ |  | 63.0 |
| ✓ | ✓ |  |  | ✓ | 63.5 |

### 4.4.1 Component-Wise Ablations

We perform ablation studies regarding each component of our CyCTR in Table 4. The first line is the result of our baseline, where we use two residual blocks to merge features as stated in Section 4.2. For all ablations in Table 4, the hidden dimension is set to 128 and two transformer encoders are used. The mIoU results are averaged over four splits. Firstly, we only use the self-alignment block that only encodes query features. The support information in this case comes from the concatenated global support feature and the prior map used in [44]. It can already bring decent results, showing that the transformer encoder is effective for modeling context for few-shot segmentation. Then, we utilize the cross-alignment block but only with the vanilla attention operation in Equation 1. The mIoU increases by 0.4%, indicating that pixel-level features from support can provide additional performance gain. By using our proposed cycle-consistent attention module, the performance can be further improved by a large margin, *i.e.* 0.6% mIoU compared to the vanilla attention. This result demonstrates our cycle-consistent attention's capability to suppress possible harmful information from support. Besides, we assume some background support features may also benefit the query segmentation and therefore use the cycle-consistent transformer to aggregate pixel-level information from background support features as well. Comparing the last two lines in Table 4, we show that our way of utilizing beneficial background pixel-level support information brings 0.5% mIoU improvement, validating our assumption and the effectiveness of our proposed cycle-consistent attention operation.

Besides, one may be curious about whether the noise can also be removed by predicting the aggregation position like the way in Equation 6 for aggregating support features to query. Therefore, we use predicted aggregation instead of the cycle-consistent attention in the cross-alignment block, as denoted by *CyCTR(pred)* in Table 4. It does benefit the few-shot segmentation by aggregating useful information from support but is 0.9% worse than the proposed cycle-consistent attention. The reason lies in the dramatically changing support images under few-shot segmentation testing. The cycle-consistency is better than the learnable way as it can globally consider the varying conditional information from both query and support.

### 4.4.2 Effect of Model Capacity

We can stack more encoders or increase the hidden dimension of encoders to increase its capacity and validate the effectiveness of our CyCTR. The results with different numbers of encoders (denoted as $L$) or hidden dimensions (denoted as $d$) are shown in Table 5a and 5b. While increasing $L$ or $d$ within a certain range, CyCTR achieves better results. We chose $L = 2$ as our default choice for accuracy-efficiency trade-off.

Table 5: Effect of varying (a) number of encoders $L$ and (b) hidden dimensions $d$. When varying $L$, $d$ is fixed to 128; while varying $d$, $L$ is fixed to 2.

| #Encoder | mIoU (%) | | #Dim | mIoU (%) |
|:---:|:---:|:---:|:---:|:---:|
| 1 | 62.4 | | 128 | 63.5 |
| 2 | 63.5 | | 256 | 64.0 |
| 3 | 63.7 | | 384 | 63.9 |
| (a) | | | (b) | |

### 4.5 Qualitative results

In Figure 4, we show some qualitative results generated by our model on Pascal-$5^i$. Our cycle-consistent attention can improve the segmentation quality by suppressing possible harmful information from support. For instance, without cycle-consistency, the model misclassifies trousers as "cow" in the first row, baby's hair as "cat" in the second row, and a fraction of mountain as "car" in the third row, while our model rectifies these part as background. However, in the first row, our CyCTR still

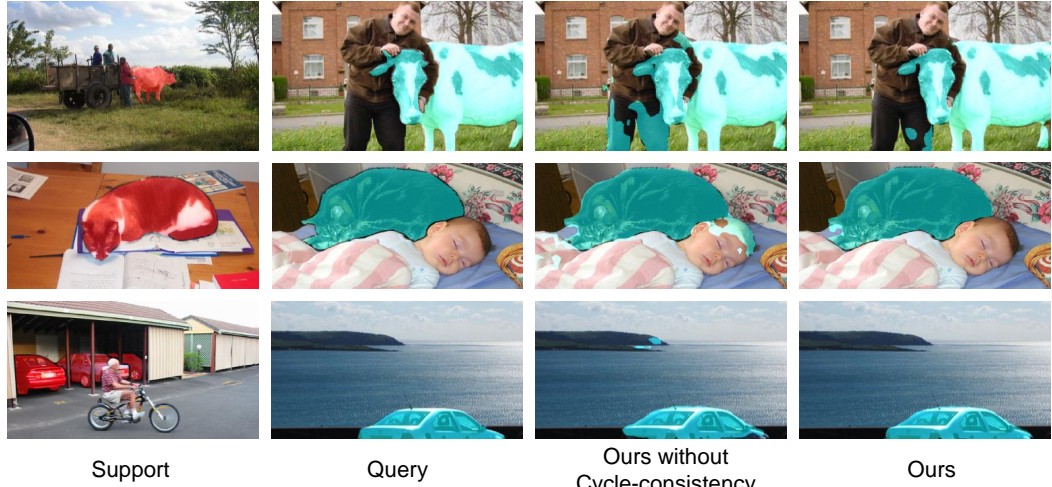

| Support | Query | Ours without Cycle-consistency | Ours |

Figure 4: Qualitative results on Pascal-$5^i$. From left to right, each column shows the examples of: Support image with mask region in red; Query image with ground truth mask region in blue; Result produced by the model without cycle-consistency in CyCTR; Result produced by our method.

segments part of the trousers as "cow" and the right boundary of the segmentation mask is slightly worse than the model without cycle-consistency. The reason comes from the extreme differences between query and support, *i.e.* the support image shows a "cattle" but the query image contains a milk cow. The cycle-consistency may over-suppress the positive region in support images. Solving such issue may be a potential direction to investigate to improve our method further.

## 5   Conclusion

In this paper, we design a CyCTR module to deal with the few-shot segmentation problem. Different from previous practices that either adopt semantic-level prototype(s) from support images or only use foreground support features to encode query features, our CyCTR utilizes all pixel-level support features and can effectively eliminate aggregating confusing and harmful support features with the proposed novel cycle-consistency attention. We conduct extensive experiments on two popular benchmarks, and our CyCTR outperforms previous state-of-the-art methods by a significant margin. We hope this work can motivate researchers to utilize pixel-level support features to design more effective algorithms to advance the few-shot segmentation research.

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
