# Supplementary Materials for
# Few-Shot Segmentation via Cycle-Consistent Transformer

## A  More Details

### A.1  Implementation

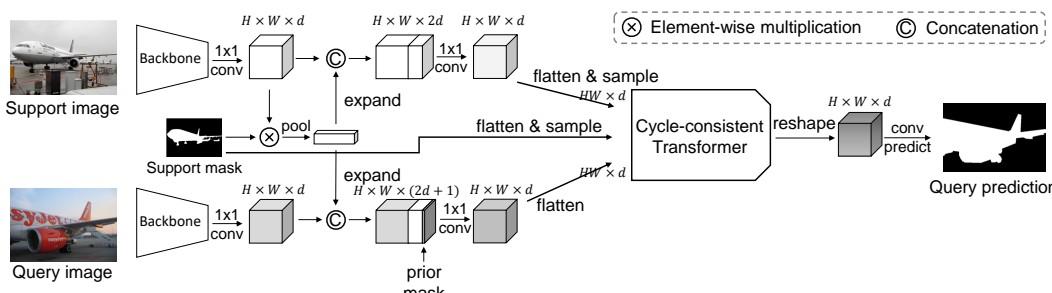

Figure 1: The network structure used in our experiments. The backbone network first extracts features for query and support images. To enable the pixel-wise comparison in transformer, the averaged foreground support feature is expanded and concatenated with both query and support features. Our Cycle-Consistent TRansformer (CyCTR) takes the flattened query and support features as well as the flattened support mask as input and produces the encoded query feature for prediction.

The overall network architecture used in our experiments is shown in Figure 1. Following the common practice [6, 7, 9], query and support image are first feed into a shared backbone network to obtain general image features. Similar to [6], the backbone network is pretrained on ImageNet [5] and then completely kept fixed during few-shot segmentation training. Following [2, 6, 8], we use dilated version of ResNet [1] as the backbone network. Besides, middle-level features are processed by a 1x1 convolution to reduce the hidden dimension and high-level features are used to generate a prior map that concatenated with the middle-level feature. In details, the middle-level feature consists of the concatenation of features from the $3^{rd}$ and the $4^{th}$ block of ResNet (total 5 blocks including the stem block) with shape $H \times W \times (512 + 1024)$ and is feed into a $1 \times 1$ convolution to reduce the dimension to $H \times W \times d$, where $d$ is the hidden dimension that can be adjusted in our experiments. The high-level feature (from the $5^{th}$ block of ResNet) with shape $H \times W \times 2048$ is used to generate the prior mask as in [6], which compute the pixel-wise similarity between the query and support high-level features and keep the maximum similarity at each pixel and normalize (using min-max normalization) the similarity map to the range of $[0, 1]$. To enable the pixel-wise comparison, we also concatenate the mask averaged support feature to both query and support feature and processed by a 1x1 convolution before inputting into the transformer. The final segmentation result is obtained by reshaping the output sequence back to spatial dimensions and predicted by a small convolution head that is consisted of one 3x3 convolution, one ReLU activation, and a 1x1 convolution. Dice loss [3] is used as the training objective.

**Baseline setup**: For the baseline of our method, we use two residual blocks [1] to merge the query feature. The support information comes from the concatenated support global feature and the prior

35th Conference on Neural Information Processing Systems (NeurIPS 2021).

map. During training, the foreground middle-level query feature from backbone network is averaged and concatenated with the middle-level support feature to predict the support mask for feature alignment. This auxiliary supervision is included in all of our experiments.

## A.2    Dataset Settings

In this Table 1 and Table 2, we provide the detailed split settings for datasets (Pascal $5^i$ and COCO-$20^i$) used in our experiments, which follow the split settings proposed in [4].

| Split | Test classes |
|---|---|
| PASCAL-$5^0$ | aeroplane, bicycle, bird, boat, bottle |
| PASCAL-$5^1$ | bus, car, cat, chair, cow |
| PASCAL-$5^2$ | diningtable, dog, horse, motorbike, person |
| PASCAL-$5^3$ | potted plant, sheep, sofa, train, tv/monitor |

Table 1: Data split for PASCAL-$5^i$, which follows the 4-fold cross-validation. Each row contains 5 classes for test and the rest 15 classes in the PASCAL dataset are used for training.

| Split | Test classes |
|---|---|
| COCO-$20^0$ | Person, Airplane, Boat, Park meter, Dog, Elephant, Backpack, Suitcase, Sports ball, Skateboard, W. glass, Spoon, Sandwich, Hot dog, Chair, D. table, Mouse, Microwave, Fridge, Scissors, |
| COCO-$20^1$ | Bicycle, Bus, T.light, Bench, Horse, Bear, Umbrella, Frisbee, Kite, Surfboard, Cup, Bowl, Orange, Pizza, Couch, Toilet, Remote, Oven, Book, Teddy, |
| COCO-$20^2$ | Car, Train, Fire H., Bird, Sheep, Zebra, Handbag, Skis, B. bat, T. racket, Fork, Banana, Broccoli, Donut, P. plant, TV, Keyboard, Toaster, Clock, Hairdrier, |
| COCO-$20^3$ | Motorcycle, Truck, Stop, Cat, Cow, Giraffe, Tie, Snowboard, B. glove, Bottle, Knife, Apple, Carrot, Cake, Bed, Laptop, Cellphone, Sink, Vase, Toothbrush, |

Table 2: Data split for COCO-$20^i$, which follows the 4-fold cross-validation. Each row contains 20 classes for test and the rest classes in the COCO dataset are used for training.

## B    More Visualizations

We provide more visualizations in Figure 2. We also provide the visualization of cycle-consistency relationships. In the first row, only a small part of the foreground region is activated while most foreground regions are valid in the second row. And in the second row, pixels on the "person" are shown in gray, which indicates that these pixels may have a negative impact on segmenting "cat".

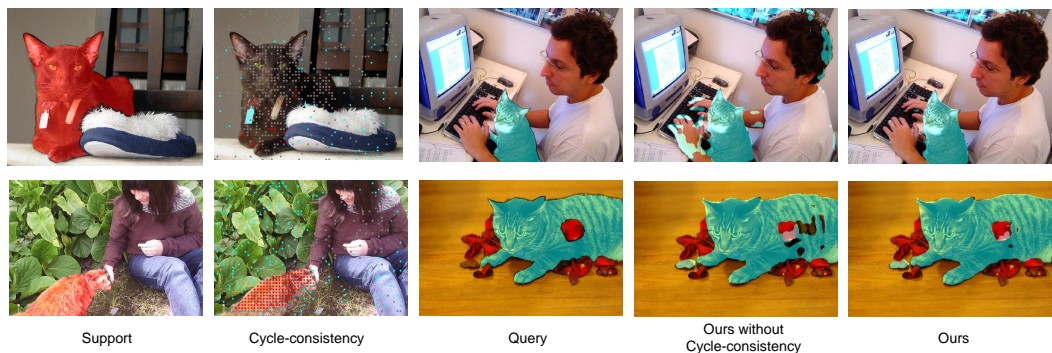

| Support | Cycle-consistency | Query | Ours without Cycle-consistency | Ours |

Figure 2: More Qualitative results on Pascal-$5^i$. The cycle-consistency is visualized in the $2^{ed}$ column, in which red points are cycle-consistent foreground pixels, blue points are cycle-consistent background pixels, and gray points are cycle-inconsistent pixels. Best viewed in color and with zoom-in.