# OpenReview forum: "Few-Shot Segmentation via Cycle-Consistent Transformer"
_NeurIPS.cc/2021/Conference — NeurIPS 2021 Poster_

### Official Review · Reviewer_asGa · 2021-07-12

**Rating:** 6
**Confidence:** 3

**Summary:**

The paper introduces a novel module called Cycle-Consistent Transformer (CyCTR) that aggregates pixel-level support features into query features via cross-attention. Cycle consistency filters out irrelevant pixel-level support features. CyCTR achieves SOTA results on Pascal-5 and COCO-20 by an appreciable margin.

**Limitations And Societal Impact:**

The paper discusses limitations at the end of the experiments section. It does not include any comments on societal impact, other than the response to check list at the very end of the submission.

**Main Review:**

The paper contains novel and interesting approach that beats existing SOTA approaches for few shot segmentation on Pascal-5 and COCO-20 by an appreciable margin. Parts of the paper have clarity while some lack clarity.

CyCTR consists of Self-alignment block and Cross-alignment block. In the former, Query, Key and Value come from the same embedding. In the latter, the Query vector comes from the query images while Key and Value come from the support image. While CyCTR uses pixel-level support features, some of its competing approaches use semantic level prototypical features from support image or the foreground region in support image to encode query features.

### Recommendation:
Section 3 (specifically its second half) has a lot of typos, poorly written and lacks clarity. Likewise, for the bottom paragraph on page 8. There is a stark difference between other sections. Recommend authors to review and improve clarity. Otherwise, the mathematical expressions are sound.


**Time Spent Reviewing:**

2

---

> ### Author Response · Authors · 2021-08-10
> **Response to Reviewer asGa**
>
> Thank you for taking the time to read our work with positive feedbacks and suggestions for improvement.
>
> We will carefully fix the typos and improve our writing in section 3 and section 4.4.1.

---

### Official Review · Reviewer_AxLz · 2021-07-14

**Rating:** 5
**Confidence:** 3

**Summary:**

The paper introduces a new transformer architecture designed to tackle image segmentation in the few-shot setting. The new architecture seems to make it easier for the pre-trained model to generalise to new classes and produce more accurate segmentation masks. The comparison with benchmarks shows incremental improvements on PASCAL-5i and COCO-20i in 1-shot and 5-shot setting, with two different convolutional backbones.

Due to minor yet numerous issues the paper is currently below the acceptance bar, but I am willing to improve my score if these are addressed.

**Limitations And Societal Impact:**

The paper does not address those issues. Negative societal impact of this work is probably similar to that of any work in machine vision - authors may want to discuss the impact of digital surveillance since improvements in image segmentation might results in greater capacity of those systems.

As for the limitations, it would be interesting to see how the number of training classes affects the scores.

**Main Review:**

The proposed method is novel and is targeted at improving the cross-attention mechanism so that inadequate keys are not attended to. Few-shot segmentation is an important problem in machine vision both application- and research-wise as fully labelled data is scarce and quick generalisation to new classes in general remains difficult for machine learning systems.

Details:
1. The proposed cycle-consistent transformer, for a given query image token, attends only to those support tokens that satisfy the cycle-consistency rule: support token i and query token j are in cycle consistency if j has the highest affinity value (QK^T attention matrix) with i among all query tokens and vice versa. This seems to be a 'hard' attention, in contrast to the softmax attention in popular transformers. Authors could potentially provide more intuition on that, for example, what happens if for a given query token there is no support one that satisfies cycle-consistency - in such a case, aren't all attention weights equal to 0, due to additive bias?
2. The method requires a vision-backbone model such as ResNet. It is mentioned that this was pre-trained, but it is unclear if that part was also fine-tuned together with CyCTR, or remained fixed? Also, the scores obtained with ResNet-101 are often worse than those with ResNet-50, in particular for 1-shot segmentation, what could that be attributed to?
3. It is unclear which 'middle-level features' and 'high-level features' (lines 233-234) are used in case of each backbone. Also it is unclear what the 'prior map' refers to.
4.  line 239 is unclear: 'As for the baseline of our method, we use two residual blocks [10] instead of the transformer to encode query feature' -  How residual blocks replace cross-attention? It should be marked in Table 4 that the first line refers to that benchmark model. Also, language errors.
5. (246) 'during test, 1000 episodes' - what are episodes here? unclear
6. Figure 4 - if the 'query' column shows the ground truth segmentation, maybe it should be marked.
7. What are the different stacked encoders that are mentioned in 4.4.2?
8. There are multiple typos, language errors and unclear formulations:
- 59: to encoder -> to encode
- 105: point -> pointed
- 113: are -> were
- 128: ideology -> idea
- 234: 'high-level feature are used to generate a prior map that concatenated with the middle level feature' - unclear
- 237: restuls -> result
- 238: convolution -> convolutional
- 238: is consisted by -> consists of
- 245: classes is used -> classes are used
- 246: during test -> at test time
- 253 training *for*  50 epochs
- 263: the number of head for all attention layers are set -> the number of heads for all attention layers is set
- 306: one may curious









**Time Spent Reviewing:**

4

---

> ### Author Response · Authors · 2021-08-10
> **Response to Reviewer AxLz**
>
> Thank you so much for acknowledging the novelty of our method. We have carefully considered your constructive and insightful comments and here are the answers to your concerns.
>
> **Q1. What happens if for a given query token there is no support one that satisfies cycle-consistency?**
>
> In the extreme case mentioned by the reviewer, weights equal to 0. In our experiments, it does exist but happens with a low frequency. If this extreme case happens all the time, the model will downgrade to the one without cycle-consistent attention, *i.e.*, the result 61.6 of the second row in Table 4. When all pixels are "cycle-consistent", it becomes the model with traditional attention, *i.e.*, the result 61.2 of the third row in Table 4. Both of them are lower than the final result of 62.8 of our model. Besides, our cycle-consistent attention is not "hard" attention since softmax is also used to calculate the attention weights within the cycle-consistent pixels.
>
> **Q2. Details about backbone networks**
>
> The pre-trained backbone is fixed during training on few-shot benchmarks, as said in Line-256.
>
> Besides, the statement by the reviewer "the scores obtained with ResNet-101 are often worse than those with ResNet-50" is actually not a common observation. Such a case only appears in previous method PFENet [1] given in Table 1. In our method, as shown in Table 1, we achieve better results with ResNet-101. DAN [2] can also obtain better results with ResNet-101. It is unknown for other methods if the mentioned phenomenon exists, because other methods only report the results with either ResNet-50 (*e.g.,* PMMs[3], PGNet[4], CANet[5]) or ResNet-101 (*e.g.,* FWB[6]).
>
> The authors of PFENet didn't give an explanation for this phenomenon in their paper [1]. We guess it is because that the extracted features from ResNet-101 are more abstract since ResNet-101 is much deeper than ResNet-50. When only using class-level features as PFENet does, the networks that take the backbone features as input would overfit on few-shot segmentation. We believe this is an open question to be explored in the future.
>
> **Q3. Explanation of details.**
>
> *1. About middle-/high-level features and prior mask.*
>
> The details are stated in the supplementary materials due to the limitation of the space.
>
> The "middle-level features" means the concatenation of features from the $3^{rd}$ and the $4^{th}$ blocks of ResNet while the high-level feature means the feature from the $5^{th}$ (last) block of ResNet backbone.
>
> The prior mask is obtained by computing the similarity between the query and support high-level features. And the calculated similarity map is concatenated with the middle-level feature and processed by a 1x1 convolution before feeding into the transformer.
>
> Note that the ways to use middle-/high-level features and the prior mask are the same as those in PFENet. We will make this clearer in our final paper.
>
> *2. About the baseline statement.*
>
> The statement "As for the baseline of our method, we use two residual blocks instead of the transformer to ..." means that our baseline is the model without the transformer blocks, and two additional residual blocks are used to process the query features for final segmentation. These two residual blocks are not used to perform cross-attention. We will clarify the statement to avoid confusion and will also mark out our baseline in Table 4 in the revision.
>
>
> *3. What are episodes here?*
>
> The "1000 episodes" represents that we evaluate our model on 1000 testing pairs/sets and average the results as the final performance. We have provided a brief explanation of episodes in Section 3.1 (Problem Setting) and will make it clearer in the experiment part to avoid confusion.
>
> *4. What are the different stacked encoders that are mentioned in 4.4.2?*
>
> Similar to the traditional transformer, we can repeat (stack) the module multiple times by taking the output of the previous one as input to the next. In 4.4.2, we ablate the number of encoders to see how this affects the final performance.
>
> **Q4. Typos**
>
> Thank you for kindly pointing out the typos. We will carefully resolve the typos.
>
> **Reference**
>
> [1] Prior Guided Feature Enrichment Network for Few-Shot Segmentation, Tian Z, et al. TPAMI, 2020.
>
> [2] Few-Shot Semantic Segmentation with Democratic Attention Networks, Wang H, et al. ECCV, 2020.
>
> [3] Prototype Mixture Models for Few-shot Semantic Segmentation, Yang B, et al. ECCV, 2020.
>
> [4] Pyramid Graph Networks With Connection Attentions for Region-Based One-Shot Semantic Segmentation, Zhang C, et al. ICCV, 2019
>
> [5] CANet: Class-Agnostic Segmentation Networks with Iterative Refinement and Attentive Few-Shot Learning, Zhang C, et al. CVPR, 2019.
>
> [6] Feature Weighting and Boosting for Few-Shot Segmentation, Nguyen K, et al. ICCV, 2019.

---

> > ### Comment · Reviewer_AxLz · 2021-08-31
> > **thank you for clarifications**
> >
> > I would like to thank reviewers for clarification on the various matters.
> >
> > One thing related to Q1 remains obscure for me: If the attention is applied only to cycle-consistent pixel pairs (i, j), and j's are selected as the ones with highest affinity to given i, isn't there usually a single j that satisfies this for given i? And hence, isn't the softmax applied to just one (or none) valid pairs, effectively reducing the whole mechanism to hard-attention? The situation where for given i we'd have multiple cycle-consistent j's seems extremely unlikely/ degenerate to me, I must be missing something and would like to understand why it is not the case.

---

> > > ### Author Response · Authors · 2021-09-01
> > > **Reply to Reviewer AxLz**
> > >
> > > We sincerely thank for the reviewer's valuable feedback. There may exist a misunderstanding of our attention mechanism and we clarify this as follows:
> > >
> > >
> > > 1) Our cycle-consistent attention can be roughly described as two steps: Firstly, we aim to discover a set of informative support pixels, **rather than the query ones**. The principle is that a support pixel is treated as valid and selected if it can find a cycle-consistent query pixel. The support pixel, which cannot build correspondence with any query pixel, will be treated as irrelevant and eliminated. Secondly, each query pixel attends to the same set of selected support pixels to aggregate useful support information.
> > >
> > > 2) The case (mentioned by the reviewer) that support token i finds a cycle-consistent query token j only means that support token i is valid and selected. But there are also other support tokens/pixels selected as informative ones. Therefore, with respect to query pixel j, the softmax operation is applied to the similarities between query j and all the selected support pixels, rather than only pair (i, j).
> > >
> > > We will make this clearer in our final version. We hope you will support the paper and consider raising the score if appropriate. If you have any further questions, we will be happy to answer.

---

### Official Review · Reviewer_1f9d · 2021-07-15

**Rating:** 6
**Confidence:** 4

**Summary:**

The paper addresses the few-shot semantic segmentation problem with cycle-consistent transformer (CyCTR) that utilizes pixel-wise support information for query prediction. The CytCTR estimate the cross-attention between support and query images with the cross-consistent attention mechanism. The proposed method achieves the start-of-the-art performance on multiple datasets.

**Limitations And Societal Impact:**

Good for me.

**Main Review:**

The idea of pixel-level cycle consistency shares similarities with [1], nonetheless, it is still interesting to see it is utilized in the attention map of the transformer modules.

1) From reviewer's understanding, when training, the attention weights of those pixels that are not “cycle-consistent” are added with a minus infinity and can be seen as disabled as they are normalized as zero after the softmax. As I do not see any explicit measures like loss function that guide the pixels to be “consistent”, this reminds me of dropout, which randomly disables a part of the network. Therefore, it is desirable to see more training details, e.g., is the percentage of pixels that are “inconsistent” decreases during the training process and reaches a low level in the late stage of the training? Any visualizations about the attention maps that can illustrate how pixels are “consistent”?

2) There may exist "misplaced mapping". For example, the nearest point of "cow" in query belongs to "horse" in support and then the nearest point of "horse" in support belongs to "cow" in query, which is actually inconsistent but will be viewed as "consistent" in this work.

3) Though the part of “Mask guided sparse sampling” shown effectiveness in the results, it lacks ablation study or comparison with other methods.

4) Some technical details are missing: e.g. when stacking L encoders, is the output of the previous encoder serves as the Z_kv of the next layer? Is X_s is inputted into each layer of the encoders? And some parts of the paper are not easy to follow: e.g. in Fig. 3, same symbols (Z_kv, Z_q) are used for inputs to two blocks (Cross-alignment, Self-alignment), which sounds like they were identical.

5) Another concern is the parameter size. The ResNet-101 is already a big model, and the transformer is also known for its resource requirements. Authors trained this few-shot model with 4 V100s, using much more resources than previous few-shot segmentation methods.

[1] Kang, Guoliang, et al. “Pixel-Level Cycle Association: A New Perspective for Domain Adaptive Semantic Segmentation.” Advances in Neural Information Processing Systems, vol. 33, 2020, pp. 3569–3580.


**Time Spent Reviewing:**

7

---

> ### Author Response · Authors · 2021-08-10
> **Response to Reviewer 1f9d**
>
> Thank you so much for acknowledging the strength of our method. We have carefully considered your constructive and insightful comments and here are the answers to your concerns.
>
> **Q1. More training details about the cycle-consistent transformer.**
>
> Our cycle-consistent attention is different from using a training loss. We aim to eliminate the inconsistent pixels when performing the attention. In this way, it benefits the feature learning by avoiding the interference from irrelevant local pixels. The percentage of inconsistent pixels would not reach a low level since we are trying to exclude these inconsistent support pixels rather than encourage them to form the cycle-consistency.
>
> Following your suggestion, we track the training process and record the proportion of cycle-inconsistent pixels in all support pixels. It gradually increases from an average ratio of 29% and converges to an average ratio of 33.5%. This may be because as the training proceeds, the features become more discriminative so that more semantically inconsistent pixels or irrelevant features can be discovered. A visualization example of the cycle consistency can be found in the supplementary materials.
>
>
>
> **Q2. The possible "misplaced mapping" cases.**
>
> We want to emphasize that our method aims at avoiding the "worst cases", *i.e.*, removing support pixels with certain inconsistency, rather than ensuring all support pixels to form cycle-consistency. Although it is impossible to remove all these semantically inconsistent support pixels without knowing the ground truth (e.g. the case mentioned by the reviewer), our way largely avoids aggregating irrelevant support features compared to the vanilla attention operation. As shown in our experiments in Section 4, compared with vanilla attention, our cycle-consistent attention does show significant improvements by not aggregating features of inconsistent support pixels. We will provide more discussions in the revised version.
>
>
>
>
> **Q3. Ablations on mask guided sparse sampling.**
>
> By default, we sample $N_s=600$ tokens for all 1-shot experiments when using the cross-alignment block. We provide the ablations with varying the number of sampled tokens below. As shown, our sparse sampling strategy consistently improves the performance compared to the method without using this sparse sampling strategy.
>
> | sampling strategy                                            | mIoU |
> | ------------------------------------------------------------ | ---- |
> | $N_s=400$                                                    | 62.4 |
> | $N_s=600$ (default)                                          | 62.8 |
> | $N_s=800$                                                    | 62.8 |
> | $N_s=1000$                                                   | 62.5 |
> | w/o sampling                                                 | 61.7 |
>
>
> **Q4. Technical details clarification.**
>
> Yes, when stacking $L$ encoders, the output of the previous encoder is fed into the self-alignment block. And $X_s$ is fed into each encoder. We will clarify the details in the revised version to distinguish symbols representing the input of different blocks (e.g. $Z_{kv}^{self}$, $Z_{q}^{self}$ for self-alignment block and $Z_{kv}^{crs}$, $Z_{q}^{crs}$ for cross-alignment block).
>
> **Q5. Concerns about computational resources.**
>
> Our method is trained with similar computational resources compared to the previous SOTA method PFENet. For experiments on COCO, we train one model on 4 GPUs which is the same as PFENet. On Pascal, the model on one split is trained with a single GPU. We say "All experiments are carried out on four Tesla V100 GPUs" as we run the experiments of 4 splits in parallel for Pascal. Besides, the experiments can also be conducted with lower-memory GPUs, *e.g.* TitanX with 12G memory. This will be made clear in the revision.

---

### Official Review · Reviewer_JH49 · 2021-07-16

**Rating:** 6
**Confidence:** 4

**Summary:**

This paper proposes a model to facilitate the few-shot semantic segmentation task by utilizing pixel-wise relationships between support and query images. The authors use a cycle-consistent attention mechanism to perform cross-attention between features from support and query images. Experiments on the two commonly used benchmarks validate the effectiveness of the proposed method.

**Limitations And Societal Impact:**

Based on my knowledge, this paper poses no negative societal impacts.

**Main Review:**

Pros:

* The idea of considering pixel-wise relationships between support and query images is interesting.
* The paper is well-written, easy to follow.
* Extensive experiments with adequate discussions.
* The proposed method achieves competitive results on two benchmarks.

Cons/Questions:

* I'm confused why the model is designed for the pixel-wise relationship. From the network structure in the supplementary, the input of the cycle-consistent transformer is the features combined with the class prototype, which is obtained by the masked average pooling method. The prototype is class-level, right? Why called the cycle-consistent transformer considers the pixel-wise relationship?
* The whole CyCTR module consists of stacking L encoders. What's the setting of L in the reported experiments?
* The method seems to be computationally a little expensive. Did the authors analyze the efficiency of the proposed method?
* Current benchmark is established on the one-way task, which is a binary mask prediction. Can the proposed method be applied to multi-way semantic segmentation tasks?

typo: L210 "... offeset from each piexl and P represents" should be pixel

**Time Spent Reviewing:**

5

---

> ### Author Response · Authors · 2021-08-10
> **Response to Reviewer JH49**
>
> Thank you so much for acknowledging the strength of our method. We have carefully considered your constructive and insightful comments and here are the answers to your concerns.
>
> **Q1. Why the model is designed for the pixel-wise relationship?**
>
> The input of the cycle-consistent transformer is pixel-wise, which provides abundant local details. Combining with the class-level features is just to introduce some global information into each pixel. Both local and global information is essential for the few-shot segmentation.
>
>
> **Q2. The setting of L encoders.**
>
> We use L=2 encoders in our experiments. We have also reported results with different L in Table 5.
>
> **Q3. Efficiency analysis.**
>
> We analyze the efficiency of our method and provide a comparison of inference speed in terms of fps (comparing with the previous SOTA PFENet using their public codes). We run both models with ResNet-50 backbone on a single Tesla V100 GPU. Note that although our method is not designed for efficiency purposes, it still has a fast inference speed.
>
> | method | mIoU | fps  |
> | ------ | ---- | ---- |
> | PFENet | 60.8 | 18.2 |
> | Ours   | 62.8 | 15.1 |
>
> **Q4. Can the proposed method be applied to multi-way semantic segmentation tasks?**
>
> Our method can be applied to multi-way segmentation by extending the model with multiple binary decoders or running the binary segmentation multiple times. Because most previous works are evaluated on these one-way benchmarks, we just follow this setting to make a fair comparison.

---

> > ### Comment · Reviewer_JH49 · 2021-09-03
> > **Keep Initial Rating**
> >
> > Thanks to the authors for the response. After careful review of feedback from all reviewers and answers by the authors, I would like to keep the initial rating.

---

### Decision · Program_Chairs · 2021-09-27

**Decision:**

Accept (Poster)

**Comment:**

This paper proposes to do few-shot segmentation by applying a transformer to the few-shot support examples. The transformer’s attention is constrained so that only pixels that meet the proposed “cycle-consistency” constraint may be used for the prediction task. While the submission isn’t particularly groundbreaking methodologically, the approach seems to be novel overall and the quantitative results are quite strong across two standard benchmarks: for example on COCO, the proposed approach outperforms previous SotA by 40% to 32% for 1-shot, or 45% to 37% for 5-shot. The method is ablated to demonstrate the benefit of various parts of the model, e.g. the use of the cycle-consistency constraint vs. a baseline transformer with no cycle-consistency masking.

Reviewers pointed out a number of clarity and presentation issues with the submission. Please take these suggestions into account and revise the text accordingly for the camera-ready. Overall the text can be a bit difficult to parse at many points, and this caused confusion among reviewers and myself -- please try to revise and clarify wherever possible to phrase things more straightforwardly, especially in the core method section.

Other few-shot segmentation papers reported similar results to those in the submission [1, 2], but as they were roughly concurrent with this submission, the submission can’t be penalized for them. However, the authors are strongly encouraged to include comparisons with these recent works in the camera-ready version, and provide context comparing their proposed approach to these methods.

Given the novelty of the approach and the strength of the results, I recommend accepting the paper to NeurIPS.

[1] https://arxiv.org/abs/2104.01538
[2] https://arxiv.org/abs/2103.15402